Journal of
open psychology data

# A Dataset of Social-Psychological and Emotional Reactions During the COVID-19 Pandemic Across Four European Countries

DATA PAPER

**DAVID ABADI** [iD]

**IRENE ARNALDO**

**AGNETA FISCHER** [iD]

*Author affiliations can be found in the back matter of this article

]u[ ubiquity press

## ABSTRACT

In April 2020, only a few weeks after the COVID-19 pandemic had erupted, we conducted an online survey and collected data from 2031 individuals in four European countries (Germany, the Netherlands, Spain and the United Kingdom) using a cross-sectional design. Participants recruited on *Cint* completed new and pre-existing measures of socio-political and populist attitudes perceived threats, appraisals (anger at the government, anger at transgressors of hygiene measures, anxiety about coronavirus via the appraisals of health-related threats), conspiracy mentality, moral reasoning, threat estimation (coronavirus, climate, symbolic material/safety), news consumption, support for and compliance with governmental hygiene measures, subjective social status and demographics. The dataset is stored on figshare repository. It can be used to study social-psychological, emotional, socio-political and socio-economic factors of the COVID-19 pandemic.

**CORRESPONDING AUTHOR:**

**David Abadi**

University of Amsterdam, NL

d.r.abadi@uva.nl

**KEYWORDS:**
anger; anxiety; appraisal theory; conspiracy mentality; COVID-19 pandemic; populism; threat

**TO CITE THIS ARTICLE:**
Abadi, D., Arnaldo, I., & Fischer, A. (2023). A Dataset of Social-Psychological and Emotional Reactions During the COVID-19 Pandemic Across Four European Countries. *Journal of Open Psychology Data,* 11: 11, pp. 1–11. DOI: https://doi.org/10.5334/jopd.86

# (1) BACKGROUND

From the start, COVID-19 has been an existential threat, affecting not only our physical health, but also our social lives and personal well-being, such as mental, emotional and social health factors. At the time of our data collection in April 2020, the early stages of the pandemic, it was still unknown how exactly the coronavirus spreads, how multiple infections could take place, and whether potential vaccines might help against its different variants. This global lack of scientific knowledge about and control over this existential threat triggers anxiety and uncertainty among people.

To date, many studies in EU countries (Mazza et al., 2020; Ozamiz-Etxebarria, Dosil-Santamaria, Picaza-Gorrochategui, & Idoiaga-Mondragon, 2020; Pieh, Budimir, & Probst, 2020; Robillard et al., 2020) have shown that people are experiencing anxiety in response to the COVID-19 pandemic. Not everyone is equally anxious, however, depending on how one estimates the risk of becoming infected (Xu & Cheng, 2021). This may explain why in some studies people with the highest probability of becoming infected are the most anxious, such as the elderly (Hyland et al, 2020), people whose friends or family have become infected, or those who lived close to pandemic hotspots. Anxiety therefore appears to vary with proximity to perceived sources of infection (see e.g., Cao et al., 2020).

The uncertainty that characterizes such threats has broader implications and brings about threats to one's self-esteem, one's goals in life, or one's social relevance, and motivates the restoration of one's sense of self and significance in life (Kruglanski et al. 2021). One of the ways this restoration can occur is through blaming others who are seen as responsible for the negative outcomes in one's life (e.g., Greenberg, Solomon, & Pyszczynski, 1997; van Prooijen, 2019). Existential threats also give rise to making sense of the threatening events. Conspiracy theories (van Prooijen & Acker, 2015) help to explain impactful events, such as a pandemic, with simplistic, one-sided, and proportionally large causes that are often the result of the deliberate will of a clandestine and powerful group, such as cults, secret organizations or extraterrestrials (Leman & Cinnirella, 2007; van Prooijen & Douglas, 2017; van Prooijen, 2019).

People differ in how susceptible they are to explanations based on conspiracy theories, which is referred to as conspiracy mentality (Bruder, Haffke, Neave, Nouripanah & Imhoff, 2013). Conspiracy mentality predicts beliefs in specific conspiracy theories and has been shown to be related to right-wing authoritarianism (Dyrendal, Kennair, & Bendixen, 2021; Imhoff, 2015). Research on the relationship between stress, anxiety, and conspiracy beliefs has shown inconsistent results, however.

Salient out-groups can also be those in positions of power, however, such as governments, politicians or CEOs (Douglas, 2021). Populism is characterized by the division of society into two homogeneous and antagonistic groups (e.g., Mudde, 2004; Wirth et al., 2016; Schulz, Müller, Schemer, Wirz, Wettstein, Wirth, 2018; Wirz, 2018). This refers to the opposition between "the pure people" and "the corrupt elites" (*Manichean dichotomy*), which has been described as the essence of populism (e.g., Mudde, 2004; Rodrik, 2020).

This antagonistic thinking can also be found in conspiracy theories, in which "actors join together in a secret agreement to try to achieve a hidden goal" (Van Prooijen et al., 2015). Indeed, previous research has also shown a positive relationship between conspiracy mentality and populist attitudes (Balta, Kaltwasser & Yagci, 2021; Castanho Silva, Vegetti, & Littvay, 2017; Hameleers, 2021) or political extremism (van Prooijen, Krouwel, & Pollet, 2015).

Applying these insights to the COVID-19 pandemic, we assume that the coronavirus causes mortality salience. Anxious individuals may exhibit two different types of responses: avoiding the threat by complying with hygiene measures or fighting the threat by showing anger at the government and adopting populist mindsets or by denying or trivializing the threat (conspiracy beliefs). There is indeed strong evidence for anger and anxiety to reinforce populist and extremist attitudes (Abadi, Bertlich, Duyvendak , & Fischer, 2023), as indicated by increased in-group favoritism and out-group hostility (Greenberg et al., 1990; Rosenblatt et al., 1989; Schimel et al., 1999), but also conspiracy thinking (e.g., Grzesiak-Feldman, 2013; Hollander, 2018; Swami & Furnham et al., 2016). In addition, anxiety has been shown to predict behavioral changes in personal hygiene and social isolation (Harper et al., 2020).

We wondered whether anxiety is associated with increased anger, either at violators of hygiene measures or at the government, as well as with support for and compliance with governmental hygiene measures, and how these are influenced by socio-political attitudes and conspiracy mentality. At the time of our data collection, the countries differed in the number of COVID-19 related deaths, as well as in the measures being taken by their respective governments, with Spain having the highest number of deaths, followed by the UK. On April 13, 2020, Spain counted more than 17,000 deaths, while the Netherlands and Germany reported less than 3,000 deaths. During the same week, the UK saw a sharp increase in coronavirus deaths, with more than 11,000 deaths counted. We selected four countries with different public health laws, socio-economic and socio-political contexts, and different implementation of lockdown and hygiene measures during the COVID-19 pandemic.

# (2) METHODS

## 2.1 STUDY DESIGN

We conducted an online survey in April 2020 in four different European countries (Germany, the Netherlands,

Spain, and the UK; N = 2031) using a cross-sectional design. The dynamics of social-psychological and emotional reactions during the COVID-19 pandemic are central to this data collection.

## Procedure

The survey was first developed in English and then translated by native speakers into three other languages before being back-translated into English, which did not reveal any major issues. In addition, each survey version was customized based on country specifications, such as country name and culture-specific terms. All translated surveys were uploaded to the *Qualtrics XM* online survey platform (version: April 2020) and respondents were recruited and survey data collected through *Cint*, which is a global research platform that provides a heterogeneous pool across all four countries involved in this study. Recruitment and incentive conditions are standardized by the *ESOMAR Questions*,[1] which are designed to guide and assist purchasers of online research samples:

## 2.2 TIME OF DATA COLLECTION

The data were collected during the initial phase of the COVID-19 pandemic, between 12 and 14 April 2020, across four European countries (Germany, the Netherlands, Spain, and the UK). By the time our survey was conducted in April 2020, Spanish citizens had already been under full lockdown for four weeks (since March 14, 2020). On March 16, the Prime Minister of the Netherlands addressed the nation to inform them about *social distancing* measures that were less strict than in other European countries ('*intelligent lockdown*'). On the same day, the state of Bavaria in Germany declared the state of emergency, followed soon after by other German states. The measures taken in Germany varied from state to state, and it is therefore difficult to draw general conclusions about the strictness of policy measures for the country as a whole. In the UK, the measures came into force on March 26, so our UK sample had already experienced the lockdown for over 2 weeks.

## 2.3 LOCATION OF DATA COLLECTION

Our country samples included Germany, the Netherlands, Spain, and the UK.

## 2.4 SAMPLING, SAMPLE AND DATA COLLECTION

### Respondents

Our desired representative sample size amounted to approximately 500 respondents per country. In the informed consent respondents were instructed about the purpose of our study, their voluntary participation and guaranteed privacy based on GDPR regulations. Our final sample consisted of 2031 participants (N = 2031, $M_{age}$ = 41.32, SD = 12.52). The characteristics of our sample

across four countries included quotas based on current UN-census data (*United Nations Data Retrieval System*), established for age, gender and geographic region (see Table 1). Specifically, quotas based on current UN-census data were set up for the following geographic regions within each country:

Germany: Baden-Württemberg, Bayern, Berlin, Brandenburg, Hessen, Thüringen, Mecklenburg-Vorpommern, Niedersachsen, Bremen, Nordrhein-Westfalen, Rheinland-Pfalz, Saarland, Sachsen, Sachsen-Anhalt, Schleswig-Holstein, Hamburg

Spain: A.M Barcelona, A.M Madrid, Centro (Central), Levante (Central East), Noreste (North East), Noroeste (North West), Nortecentro (North Central), Sur (South)

The Netherlands: Eastern Netherlands, Northern Netherlands, Southern Netherlands, Western Netherlands

United Kingdom: East of England, London, Midlands, North East Yorkshire & the Humber, North West, Northern Ireland, Scotland, South East, South West, Wales

## 2.5 MATERIALS/SURVEY INSTRUMENTS

**Measures.** Our dataset includes a variety of measures, covering social-psychological, emotional, socio-political, socio-economic and demographic variables. To this end, we included measures that capture appraisals (anger at the government, anxiety about coronavirus via the appraisals of health-related threats), perceived threats and threat estimation (coronavirus, climate, symbolic material/safety). Although anxiety and threat related to COVID-19 are primarily health-related, they also have other implications. Therefore, we also included measures related to socio-political attitudes and socio-economic factors, such as political orientation, populist attitudes, conspiracy mentality, moral reasoning, news consumption, hygiene measures, subjective social status and demographics. In short, we included measures of broad interest to social scientists and with a long tradition in the social-psychological literature. See Table 2 for all instruments, items, response options, mean, standard deviation and reliability (Cronbach's alpha).

## Demographics

We collected self-reported data on gender, employment and marital status.

## Subjective Social Status

We used the *MacArthur Scale of Subjective Social Status* (Adler, Epel, Castellazzo, & Ickovics, 2000), which

| VARIABLES | CATEGORIES | GERMANY (N = 524) | SPAIN (N = 496) | NETHERLANDS (N = 503) | UK (N = 508) |
|---|---|---|---|---|---|
| Age (%) | 18–24 | 8.97 | 9.88 | 9.74 | 12.01 |
| | 25–34 | 22.14 | 22.98 | 22.66 | 23.82 |
| | 35–44 | 23.09 | 34.48 | 25.85 | 23.82 |
| | 45–54 | 26.34 | 19.56 | 21.47 | 21.46 |
| | 55–64 | 17.75 | 12.1 | 18.29 | 17.32 |
| | 65–74 | 1.72 | 1.01 | 1.79 | 1.58 |
| | 75–84 | 0 | 0 | 0.199 | 0 |
| Gender (%) | Male | 50.76 | 50.2 | 52.49 | 47.44 |
| | Female | 49.05 | 49.8 | 47.52 | 52.56 |
| | Other | 0.191 | 0 | 0 | 0 |
| Employment (%) | Unemployed | 11.26 | 14.11 | 20.08 | 15.75 |
| | Student | 6.3 | 6.05 | 5.96 | 2.95 |
| | Retired | 8.02 | 2.22 | 2.78 | 3.15 |
| | (Self-)Employed | 74.43 | 77.62 | 71.17 | 78.15 |
| Education (%) | No degree | 2.1 | 0.61 | 1.79 | 5.12 |
| | High school | 11.64 | 15.52 | 14.51 | 22.05 |
| | Some university, no degree | 8.59 | 5.65 | 36.18 | 14.76 |
| | Technical degree | 46.18 | 23.79 | 22.47 | 18.11 |
| | Bachelor's degree | 13.36 | 38.11 | 8.95 | 26.97 |
| | Master's degree | 16.79 | 11.9 | 12.33 | 8.47 |
| | Doctoral degree | 1.34 | 4.44 | 3.78 | 4.53 |
| Religion (%) | Protestant | 23.86 | 1.82 | 12.33 | 19.09 |
| | Roman-Catholic | 25.76 | 46.17 | 20.48 | 19.49 |
| | Muslim | 5.73 | 0.81 | 5.77 | 4.73 |
| | Jewish | 0.76 | 0.61 | 1.59 | 1.38 |
| | Russian-Orthodox | 0.95 | 0.4 | 0.4 | 0.4 |
| | Greek-Orthodox | 0.76 | 0 | 0.4 | 0.79 |
| | Hindu | 0.76 | 0 | 0.4 | 0.98 |
| | Buddhist | 0.57 | 0.61 | 0.99 | 0.79 |
| | Agnostic | 1.91 | 6.25 | 0.99 | 1.77 |
| | Atheist | 6.68 | 14.52 | 3.58 | 5.91 |
| | Spiritual | 1.91 | 4.64 | 4.97 | 3.35 |
| | Non-Religious | 30.34 | 24.19 | 48.11 | 41.34 |

**Table 1** Demographic Variables across four countries (N = 2031).

represents an ascending ladder from zero to ten and measures the socio-economic status as subjectively perceived by respondents.

### Political Orientation

We asked participants to place their political orientation on a spectrum between left-wing (–5) and right-wing (+5).

**Anxiety about Coronavirus**. The object of anxiety or fear can be an event, a person, a social group, a symbol that people are afraid of, or 'the system' as a very diffuse notion of how the powerful few protect their interests. The appraisal of threat has been described as the "core relational theme" of anxiety (Smith & Lazarus, 1993), implying that threat is the defining characteristic of anxiety experiences. We developed this scale to measure

| INSTRUMENT | NUMBER OF INCLUDED ITEMS (*REVERSED) | NUMBER OF RESPONSE OPTIONS | MEAN | SD | RELIABILITY (CRONBACH'S ALPHA) |
|---|---|---|---|---|---|
| Demographics | 1 | 8 | 41.32 | 12.52 | N/A |
| Subjective Social Status | 1 | 1 | 6.048 | 1.829 | N/A |
| Political Orientation | 1 | 11 | −0.042 | 2.188 | N/A |
| Anxiety about Coronavirus | 5 | 10 | 6.197 | 1.433 | 0.554 |
| Anxiety about Coronavirus (items removed) | 4 | 10 | 6.861 | 1.958 | 0.808 |
| Feeling Status | 1 | 10 | 6.195 | 2.08 | N/A |
| Coronavirus Infection Status (Self and Friends) | 1 | 4 | 1.855 | 0.518 | N/A |
| Coronavirus Infection Status (Self and Friends) | 1 | 4 | 2.402 | 0.881 | N/A |
| Threats (Realistic/Symbolic) | 8 | 7 | 3.656 | 0.844 | 0.557 |
| Threats (Realistic/Symbolic) (items removed) | 7 | 7 | 3.517 | 1.037 | 0.635 |
| Threat Estimation (Coronavirus) | 6 | 10 | 6.269 | 1.579 | 0.65 |
| Threat Estimation (Climate, Symbolic Material and Safety) | 12 | 10 | 5.431 | 1.349 | 0.797 |
| Populist Attitudes | 10 | 7 | 4.618 | 0.78 | 0.676 |
| Conspiracy Mentality | 6 | 7 | 4.344 | 1.102 | 0.788 |
| Religion vs. Spirituality | 1 | 10 | 3.956 | 2.805 | N/A |
| Religion vs. Spirituality | 1 | 10 | 4.657 | 2.841 | N/A |
| Anger at the Government | 7 | 7 | 3.928 | 1.192 | 0.809 |
| Anger at Transgressors | 4 | 7 | 4.948 | 1.19 | 0.724 |
| Hygiene Measures (General) | 9 | 7 | 5.789 | 0.481 | 0.878 |
| Hygiene Measures (Personal) | 9 | 7 | 4.67 | 1.311 | 0.78 |
| Civil/Privacy rights | 1 | 10 | 4.435 | 2.793 | N/A |
| Civil/Privacy rights | 1 | 10 | 5.477 | 2.913 | N/A |
| Prosocial Behavior | 4 | 7 | 2.405 | 0.478 | 0.832 |
| Moral Reasoning I (Judgment) | 6 | 7 | 4.486 | 1.468 | 0.842 |
| Moral Reasoning II (Reaction) | 5 | 7 | 3.569 | 1.125 | 0.554 |
| Moral Reasoning II (Reaction) (items removed) | 3 | 7 | 3.332 | 1.472 | 0.704 |
| Moral Reasoning III (Values) | 11 | 7 | 5.295 | 0.817 | 0.775 |

**Table 2** Instruments, Number of Included Items and Response Options, Means, Standard Deviations and Reliability Scores (Cronbach's Alpha).

anxiety about coronavirus via the appraisals of health-related threats, which included five items.

**Feeling Status**. Here, we were interested in how the participants currently feel from negative (1) to positive (10).

**Coronavirus Infection Status**. Here, we asked whether the participants and their social environment (friends or family) were infected with the coronavirus.

### Threats (Realistic/Symbolic)

This scale was based on eight existing items, measuring *realistic* and *symbolic* threats (Stephan, Ybarra, &

Morrison, 2009) that reflect participants' anxiety about their personal well-being, both in terms of economic conditions, and cultural identity.

### Selection of News Headlines

Here, we were interested in how respondents cognitively processed a variety of news headlines (four experimental conditions; independent variables or predictors) and how their selection predicted anxiety about coronavirus (dependent variable), for example. For this purpose, we created a within-subjects (repeated-

measures) experimental design in which all respondents experienced the same conditions (coronavirus vs. non-threatening; threatening vs. non-threatening; coronavirus vs. threatening; conspiracy vs. non-conspiracy). We integrated our design into a cover story about what topics respondents would prefer to read.

## Threat Estimation

We created various items to allow respondents to estimate their perceived level of threat (i.e. appraisals) in the context of coronavirus as well as symbolic threats related to material and safety, which included items related to environmental and immigration threat scenarios.

## Populist Attitudes

This scale was based on existing items measuring populist attitudes (Akkerman, Mudde, and Zaslove, 2014; Schulz et al., 2018), and it was recently revised by Castanho Silva, Jungkunz, Helbling and Littvay (2019). This scale consists of three sub-scales, i.e. *people-centrism* (e.g., "Politicians should always listen closely to the problems of the people", *anti-elitism* (e.g., "The government is pretty much run by a few big interests looking out for themselves"), and *Manichaean outlook* (e.g., "You can tell if a person is good or bad if you know their political views").

## Nativism

Nativism focuses on the idea that people who are native to a country believe they have more rights to be treated fairly, and to receive preferential treatment when living in their country of birth (Hochschild, 2018). One previous attempt to measure nativism is the *Ipsos Nativism Scale* (Young, 2016; Zhao, 2019). The scale captures anti-immigrant perceptions, describing foreigners as taking away jobs and social services from the 'native' population and thereby weakening the economy. We found these items too constrictive for our research as they did not cover other relevant issues. We therefore created a new scale with three items to measure *nativism*, covering important topics, such as housing market, identity, culture and values, such as "The political elites have failed to protect our cultural identity".

**Conspiracy Mentality**. This scale included five items from the existing scale Conspiracy Mentality Questionnaire (CMQ; Bruder, Haffke, Neave, Nouripanah & Imhoff, 2013), such as " I think there are secret organizations that greatly influence political decisions". Given the long history of pandemics inciting anti-Semitism and its recent resurgence (see Brackmann, 2020; Gerstenfeld, 2020; Kofta, Soral & Bilewicz, 2020), we decided to include the item "Jews or Zionists have engineered the coronavirus as a biological weapon, in order to dominate the world".

**Anger at the Government**. We developed this scale to measure how respondents felt about their government's recent actions concerning the COVID-19 pandemic. It included seven items based on previous research on anger, measuring the most important anger appraisals (on a 7-point Likert-scale from strongly disagree to strongly agree), for example "I think that our government can be blamed for not reacting fast enough to the outbreak of the coronavirus".

**Anger at Transgressors**. We developed this scale to measure how angry respondents were when other people violated the hygiene measures set by the government during the COVID-19 pandemic. It included four items (on a 7-point Likert-scale from strongly disagree to strongly agree), such as "I think that the main problem is that some people do not follow the rules", or "I would confront people who transgress the rules".

**Support for Hygiene Measures**. This scale was designed to assess the level of agreement with various hygiene measures imposed during the pandemic. The scale included nine items (on a 7-point Likert-scale from strongly disagree to strongly agree), such as " Hand washing for 20 seconds more than 5 times a day" and " Wearing a face mask when leaving your house".

**Compliance with Hygiene Measures**. This scale included the same nine items as Support for Hygiene Measures, but asked respondents to indicate the extent to which they themselves adhere to these hygiene measures themselves (on a 7-point Likert-scale).

## Civil/Privacy rights

Here, we were interested in the extent to which the respondents perceived these unprecedented governmental hygiene measures as a violation of their civil and privacy rights.

## Prosocial Behavior

We created four items to measure prosocial behavior, which is defined as behavior that involves costs for the self and results in benefits to others (Wittek, & Bekkers, 2015). Prosocial behavior can benefit both the recipient and the prosocial person (mutualism) or it can only benefit others with a net cost to the person who engages in it (altruism). Here, we were interested in whether respondents would engage in prosocial behavior during a pandemic and its associated existential threat.

## Moral Reasoning

We created these items to test whether the respondent's anxiety about coronavirus predicts different types of moral reasoning (judgment, reaction, values) about their fellow citizens during the pandemic. Regarding values, we used some items from the Moral Foundations Questionnaire MFQ (Graham et al., 2011), a measure that assesses priorities in five foundational

                                                                                           

domains of moral decision-making: harm/care, fairness/reciprocity, ingroup/loyalty, authority/respect, and purity/sanctity.

## 2.6 QUALITY CONTROL

A *pre-test* (pilot study) was conducted with 50 respondents per country to evaluate the time required to complete the survey (on average between 15 and 20 minutes). It also aimed to assess the clarity of the survey items and their suitability for respondents in different countries. Our pre-test results were satisfactory and no further revisions to the survey were required. We added an attention check question (see item 18.1), and none of the participants failed this question, indicating that the participants were focused and reliable.

In total, our survey yielded 2062 respondents, while 31 respondents with missing values (unanswered questions) were excluded, resulting in 2031 complete respondents across four European countries. In addition, we also double-checked the data for any salient outliers, by assessing whether the length of interview (LOI) for all respondents was within a realistic completion time. Furthermore, we examined the standardized values (Z-scores) for univariate outliers and none exceeded the absolute value of $\pm 3.29$, which is the general cut-off level. For multivariate outliers, we obtained the scores for discrepancy (studentized residuals), leverage (hat values) and influence (Cook's distance) and they were below the critical values.

We used quotas based on UN-census data for age, gender and geographic region. In addition, our demographic data also show variability in employment status and education levels, so we obtained a relatively representative sample. Second, we used self-reports to assess people's beliefs, feelings and behaviors. While using self-reports to measure beliefs and feelings is very common in psychological research, measuring behavior through self-reports may be less accurate and more susceptible to social desirability effects than, for example, observing behavior. However, we were not aware of better methods for collecting this type of information during the COVID-19 pandemic lockdown, when people were requested to stay mostly at home.

*Cint* is the industry's largest sample exchange and the quality of its participant pool is maintained through high standards and multiple measures. *Cint* implements a cutting edge AI-driven fraud detection system that uses personally identifiable information (PII) and profiling data to detect fraudulent anomalies, such as user "surges" involving specific demographics, IP ranges or completion time frames. They also use digital fingerprinting to prevent duplicates and analyze data points to detect inattentive or disengaged behavior among panelists. Secure APIs and unique survey URLs limit common sources of fraud. A combination of proprietary *Cint Fraud Detection Services* and industry-

standard third-party solutions – including GEO IP check and CAPTCHA – are applied to ensure data quality is maintained at optimum levels. Further details about *Cint* and their commitment to quality can be found here under *Cint Quality Charter*.[2]

## 2.7 DATA ANONYMISATION AND ETHICAL ISSUES

In the informed consent, respondents were instructed about the purpose of our study, their voluntary participation and guaranteed privacy based on GDPR regulations. The studies involving human participants were reviewed, and we obtained ethical approval from the Faculty Ethics Review Board of the University of Amsterdam (number 2020-SP-12035). Patients/participants provided their written informed consent to participate in this study.

Personal data such as surname, personal identity code, address of the participant, etc., were not collected and stored in our study. All participants were assigned a unique ID and their data were stored under that ID in the dataset. Contact information for research participants was not collected, so the dataset used for analyses is completely anonymous. Participant information was anonymized by using unique identifiers for both scientific research and statistical purposes. *Cint* did not provide us any personal data and complied with the GDPR. Any personal data collected and processed by us were collected and processed independently of the services provided by *Cint*. *Cint* only provided us anonymized data that did not contain any personal data. Unless specific measures are taken to collect personal data as part of a research project or related activity, nobody will be able to determine the identity of the parties involved. More details about *Cint* and their European GDPR compliance can be found online.[3]

## 2.8 EXISTING USE OF DATA

Some measures of this dataset were used in a published study that can be accessed online here:

Abadi, D., Arnaldo, I., & Fischer, A. (2021). Anxious and Angry: Emotional Responses to the COVID-19 Threat. *Frontiers in Psychology*, 3516. https://doi.org/10.3389/fpsyg.2021.676116.

## (3) DATASET DESCRIPTION AND ACCESS

### 3.1 REPOSITORY LOCATION

The dataset presented in this article is publicly available, based on the GDPR agreements of our H2020 funded research project. The files are accessible via our *figshare* repository https://doi.org/10.21942/uva.1708 5719.[4]

### 3.2 OBJECT/FILE NAME

Abadi et al. (2023) A Dataset of Social-Psychological and Emotional Reactions during the COVID-19 Pandemic across Four European Countries.csv.

Abadi et al. (2023) Qualtrics Codebook. A Dataset of Social-Psychological and Emotional Reactions during the COVID-19 Pandemic across Four European Countries.pdf.

### 3.3 DATA TYPE

Processed data, online survey items

### 3.4 FORMAT NAMES AND VERSIONS

.csv, .pdf

### 3.5 LANGUAGE

The dataset and the corresponding files are stored in English. As the data were collected in multiple countries, the survey is also available in each translated language (Dutch, German, Spanish) upon request.

### 3.6 LICENSE

Our dataset is published under the CC BY-SA license, which allows re-users to distribute, remix, adapt, and build upon the material in any medium or format, as long the creator is credited. The license allows for commercial use. If you remix, adapt, or build upon the material, you must license the modified material under identical terms.[5]

### 3.7 LIMITS TO SHARING

The dataset presented in this article is publicly available based on the GDPR agreements of our H2020 funded research project.

### 3.8 PUBLICATION DATE

The repository on *figshare* was created on 14/04/2023 and updated on 05/07/2023.

## (4) REUSE POTENTIAL

Our dataset documented a historical event and moving targets during the unpredictable events of the COVID-19 pandemic. We included pre-existing measures that previous research has shown to be relevant, while creating new measures that we found intriguing in the context of a global pandemic. In general, our dataset can be reused to test hypotheses about social-psychological, emotional, socio-political and socio-economic factors and their interactions. In particular, these interactions can be further tested to see if they are associated with political orientation (left-wing, centrist, right-wing), subjective social status, feeling status and coronavirus infection status. For example, the following hypotheses and interactions can be tested with the measures in our dataset:

1. Appraisals (anger at the government, anger at transgressors of hygiene measures or anxiety about coronavirus) are associated with various types of threats (symbolic, realistic, economic, immigration or cultural identity).
2. Appraisals (anger at the government, anger at transgressors of hygiene measures or anxiety about coronavirus) will increase attention to any threatening information (selection of news headlines).
3. Appraisals (anger at the government, anger at transgressors of hygiene measures or anxiety about coronavirus) will lead to overestimation of various threats related to climate, symbolic or material and safety (threat estimation questions).
4. Appraisals (anger at the government, anger at transgressors of hygiene measures or anxiety about coronavirus) will lead to more support for and compliance with governmental hygiene measures (moderated by political orientation, subjective social status, feeling status or coronavirus infection status).
5. Appraisals (anger at the government, anger at transgressors of hygiene measures or anxiety about coronavirus) are positively related to socio-political factors (populist attitudes, conspiracy mentality or political orientation).
6. Appraisals (anger at the government, anger at transgressors of hygiene measures or anxiety about coronavirus) are negatively related to prosocial behavior (moderated by political orientation, subjective social status, feeling status or coronavirus infection status).
7. Appraisals (anger at the government, anger at transgressors of hygiene measures or anxiety about coronavirus) are positively related to moral reasoning (moderated by political orientation, subjective social status, feeling status or coronavirus infection status).

## NOTES

1   https://www.cint.com/esomar28.
2   https://www.cint.com/quality.
3   https://www.cint.com/gdpr.
4   https://doi.org/10.21942/uva.17085719.
5   https://creativecommons.org/licenses/by-sa/4.0.
6   https://doi.org/10.3030/822590.

## ADDITIONAL FILE

The additional file for this article can be found as follows:

• **Qualtrics Codebook.** A Dataset of Social-Psychological and Emotional Reactions during the COVID-19 Pandemic across Four European Countries.pdf. DOI: https://doi.org/10.5334/jopd.86.s1

## ACKNOWLEDGEMENTS

We thank the consortium partners of our H2020 research project for their critical feedback on our socio-political measures.

## FUNDING INFORMATION

This dataset was funded by the European Union's H2020 project *Democratic Efficacy and the Varieties of Populism in Europe* (DEMOS)[6] under H2020-EU.3.6.1.1. and H2020-EU.3.6.1.2. (grant agreement ID: 822590).

## COMPETING INTERESTS

The authors have no competing interests to declare.

## AUTHOR CONTRIBUTIONS

1. Designed study, collected data, cleaned data, documented data, edited paper and wrote paper.
2. Designed study and edited paper.
3. Designed study and edited paper.

## AUTHOR AFFILIATIONS

**David Abadi** orcid.org/0000-0001-7226-8100
Department of Psychology, University of Amsterdam, NL

**Irene Arnaldo**
Department of Psychology, University of Amsterdam, NL

**Agneta Fischer** orcid.org/0000-0001-6939-8174
Department of Psychology, University of Amsterdam, NL

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

## PEER REVIEW COMMENTS

*Journal of Open Psychology Data* has blind peer review, which is unblinded upon article acceptance. The editorial history of this article can be downloaded here:

- **PR File 1.** Peer Review History. DOI: https://doi.org/10.5334/jopd.86.pr1

**TO CITE THIS ARTICLE:**
Abadi, D., Arnaldo, I., & Fischer, A. (2023). A Dataset of Social-Psychological and Emotional Reactions During the COVID-19 Pandemic Across Four European Countries. *Journal of Open Psychology Data,* 11: 11, pp. 1–11. DOI: https://doi.org/10.5334/jopd.86

**Submitted:** 19 January 2023   **Accepted:** 28 June 2023   **Published:** 11 July 2023

