## [Peer Review History. · Journal of Open Psychology Data]

Major revisions:

Please clarify the conditions under which your data will be available, including licensing and publication time point.

Commented [A1]: On figshare, we changed the license to Creative Commons Attribution 4.0 International (CC-BY-4.0). As long as our dataset is under review the dataset files remain private.

Please provide a comprehensive codebook for your data, which at least includes a variable name, variable labels and value labels for each variable. Additionally, and in accordance with the FAIR principles data as well as any additional files (e.g. codebook) should be provided in non-proprietary formats.

Commented [A2]: The codebook was revised extensively and a separate table with our measures, items, descriptive statistics and reliability included.

Commented [A3]: The revised codebook is uploaded in PDF format.

Minor revisions:

Please revise the section 2.5 in accordance with the suggestions of Reviewer 1. In particular, I would like to ask you to insert the suggested table. Probably you could also include some reliability indicators for each scale.

Commented [A4]: Revised.

Commented [A5]: A separate table with all measures, items, descriptive statistics and reliability scores was included.

Please refer to tables 1 and 2 in the text or delete them as proposed by Reviewer 1.

Commented [A6]: Tables 1 and 2 were redundant indeed and therefore omitted.

Please provide more detailed information on the specific sample sizes in the different countries (see also Reviewer 2).

Commented [A7]: Included.

Please describe the methods used for outlier detection in more detail.

Commented [A8]: As previously described in our manuscript, the high quality of Cint's participant pool is ensured by their advanced technologies and methods. In addition, we detected salient outliers by assessing whether the length of interview (LOI) for all respondents were within a realistic completion time. Furthermore, we examined the standardized values (Z-scores) for univariate outliers and none exceeded the absolute value of ± 3.29, which is the general cut-off level. For multivariate outliers, we obtained the scores for discrepancy (studentized residuals), leverage (hat values) and influence (Cook's distance) and they were below the critical values.

The authors should consider revising the first two sentences in Section 4, "Reuse Potential." Currently, these sentences seem to depict rather general items that are not too strongly related to the present dataset.

Commented [A9]: Revised.

Comments to the author(s) *

These comments will be published along with the article itself. Reviewers are encouraged to sign their review, although anonymity is allowed.

If uploading a version of the article with tracked changes, please include a summary of your comments in the box below. This summary will be published with the article itself.

The manuscript presents a dataset (N = 2,031) including responses to various instruments on social- psychological constructs measured in four European countries at the beginning of the corona pandemic. It seems that the dataset can provide rich information on the psychological effects of the pandemic and, thus, has great reuse potential. Nevertheless, I cannot recommend its publication in its current form. I will summarize my main concerns below.

Commented [A10]: Originally, we had used the restrictive license template (RLT) as recommended to us by figshare and the data stewards of our department. This was previously explained by us but led to a misunderstanding.

Only for the purpose of review process, our dataset is privately stored on figshare, and it is now using the license Creative Commons Attribution 4.0 International (CC-BY-4.0).

The embargo is only meant for the review process and once the manuscript is published, it will be removed.

Major issues =====

1) The journal aims to publish datasets that are openly available for reanalyses by other researchers. However, if I am not mistaken this is not the case for the present dataset. The authors report that the dataset is *not* publicly available but only available to the research team. Rather, there is an unspecified embargo period on the data. Details on this embargo period are not given in the manuscript. But, the figshare repository lists a "retention period" of 2031-11-26. Does this mean the authors plan to publish the data in 8(!) years? I do understand that embargos of maybe 1 year are sometimes necessary. But I do not see the usefulness of a data paper that will not be available for a long time.

Commented [A11]: No, retention period describes the time frame until which the data will be stored on figshare. Embargo refers to the period the data remains private and unpublished.

2) The documentation of the data needs to be improved. The figshare repository includes an excel file with the raw data and PDF as a codebook. However, it is rather difficult to connect the two because the codebook does not specifically refer to the variable names in the excel file. It seems to me the numbering in the codebook refers to the variable names. But then several variables are not documented at all (e.g., A3.8, A3.10) or inadequately documented (e.g., the coding of variable A18.1). On the other hand, some variables listed in the codebook (e.g., Item 16.2) are missing from the data. I recommend revising the documentation to describe all variables in the dataset.

Commented [A12]: The codebook was revised extensively. Now it includes an introduction, in which variable names, variable labels, value labels, reversed items, intro items, attention check, country codes and RespondentID are explained.

Commented [A13]: Revised.

Commented [A14]: Attention check is explained.

Commented [A15]: The dataset was revised, duplicates and redundant data removed and missing data added.

Minor issues =====

3) The reason why the administered instruments were included in the survey is not part of the study design (page 2). I believe this information is better placed in the instrument section.

Commented [A16]: Revised.

4) What does it mean that the survey was “synchronized with a global research platform” (page 3)? I would also hope for a short description of the participant pool. How were the participants recruited and incentivized?

Commented [A17]: Revised.

Commented [A18]: The participant pool is described in sections 2.1 Study Design (Procedure) and 2.6 Quality Control. Moreover, advanced technologies and methods used by Cint to maximize the quality of their participant pool are explained.

4) The quotas for the regions (pages 3/4) should be part of the description of the sampling plan.

Commented [A19]: Revised.

5) I was wondering whether Tables 1 and 2 are necessary. More detailed information on the respondents’ age is already given in Table 3.

Commented [A20]: Tables 1 and 2 were redundant and therefore omitted.

6) It might be informative to include a table that lists all administered instruments, the number of included items, and the number of response options. This could give a nice overview of the administered measures. Moreover, the table should also include the means, standard deviations, and reliabilities (for multi-item scales).

Commented [A21]: Revised. A separate table with all measures, items, descriptive statistics and reliability scores was included.

Reviewer C:
Recommendation: Decline Submission

Comments to the author(s) *

These comments will be published along with the article itself. Reviewers are encouraged to sign their review, although anonymity is allowed.

If uploading a version of the article with tracked changes, please include a summary of your comments in the box below. This summary will be published with the article itself.

Notes:

- restrictive license template (RLT) – I don't know what this is? Seems weird to submit to open data with a restrictive license? Hard to know since this isn't explained.
 - Can you list how many participants are from each country?
 - The formatting here is odd, but I'll let the journal handle how they want to deal with that.
 - ...the mean age is 3? What is going on here? The table provides more information, but this is not calculated in a way that makes any sense.
 - Please format these tables consistently. Mention them in the text.
 - The method section could be formatted better. The question numbers don't mean anything to anyone but you.
 - Include reliability information about each scale.
 - ... do you have a codebook for this data?
 - I suppose the question numbers in the doc match the excel doc but they still aren't quite right ... a codebook / metadata would be much more helpful than these large blocks of text.
- All JOPD data papers are peer reviewed according to the following criteria:
1. The paper contents
 - a. The methods section of the paper must provide sufficient detail that a reader can understand how the dataset was created and would within reason be able to recreate it. So - So
 - b. The dataset must be correctly described. Formatting is poor and needs a codebook / metadata.
 - c. The reuse section must provide concrete and useful suggestions for reuse of the data. Seems ok
 2. The deposited data
 - a. The repository the data is deposited in must be suitable for this subject and have a sustainability model (see our list of recommended repositories).
This link doesn't work but I think figshare is ok.
 - b. The data must be deposited under an open license that permits unrestricted access (e.g. CC0, CC-BY).
Data is not open.
 - c. The deposited data must include a version that is in an open, non-proprietary format. Data is in a proprietary format. Should be converted to csv.
 - d. The deposited data must have been labelled in such a way that a 3rd party can make sense of it (e.g. sensible column headers, descriptions in a readme text file).
Data does not meet this standard.
 - e. The deposited data must be actionable – i.e. if a specific script or software is needed to interpret it, this should also be archived and accessible.
NA
 - f. Studies involving human subjects should adhere to local ethical standards at the host institution and follow American Psychological Association's (APA) Ethical Principles of Psychologists and Code of Conduct (<http://www.apa.org/ethics/code/index.aspx>). Participant data should be sufficiently anonymized and appropriate consent forms should be signed. Appears to be ok.

Commented [A22]: Originally, we had used the restrictive license template (RLT) as recommended to us by figshare and the data stewards of our department. This was previously explained by us but led to a misunderstanding.

Only for the purpose of review process, our dataset is privately stored on *figshare*, and it is now using the license Creative Commons Attribution 4.0 International (CC-BY-4.0).

Commented [A23]: Revised.

Commented [A24]: Some variables (demographics) were coded as grouped data, for which the means, SD and variance for should be calculated differently. This was revised for Age.

Commented [A25]: Revised.

Commented [A26]: Revised. A separate table with all measures, items, descriptive statistics and reliability scores was included.

Commented [A27]: Revised.

Commented [A28]: The codebook was revised.

Commented [A29]: The figshare URL remains the same and is functioning. The data files were updated according to our major revisions.

Commented [A30]: Data is stored on figshare and is now using the license Creative Commons Attribution 4.0 International (CC-BY-4.0).

Commented [A31]: Revised. Data was uploaded in Excel format and is now available in CSV format.

Commented [A32]: Revised.